# Work, Motherhood, and Nutrition: Investigating the Association of Maternal Employment on Child Nutritional Status in South Asia—A Systematic Review

**DOI:** 10.3390/nu17061059

**Published:** 2025-03-18

**Authors:** Rupali Tushar Waghode, Surabhi Singh Yadav, Ravindra Ghooi, Shariza Abdul Razak, Kavitha Chandrasekhara Menon

**Affiliations:** 1Symbiosis Institute of Culinary Arts and Nutritional Sciences, Symbiosis International (Deemed University), Lavale, Pune 412115, India; dr.rupali.waghode@gmail.com (R.T.W.); surabhi201182@gmail.com (S.S.Y.); 2Scientia Clinical Services, Pune 411013, India; ravindra.ghooi@gmail.com; 3Dietetics Programme, School of Health Sciences, Kota Bharu 16150, Kelantan, Malaysia

**Keywords:** mothers’ employment, nutritional status, children, South Asia, systematic review, malnutrition

## Abstract

Background/Objectives: Malnutrition in children is a challenge in South Asian countries, and understanding its relation with variety of social and economic conditions is imperative. The present systematic review examined the association between maternal employment and the nutritional status of children up to twelve years old from South Asia. Methods: An electronic search for research articles published in the English language between January 2011 and December 2024 was conducted in PubMed, Science Direct, and Web of Science databases. Results: A total of 10,247 articles from five South Asian countries were screened, resulting in the inclusion of 19 studies in the review based on well-defined inclusion and exclusion criteria. The findings showed that the association between maternal employment and children’s nutritional status was context-dependent, with adverse effects in children (stunting, wasting, and growth failure) when mothers worked in unskilled sectors—in low-paying jobs. Children of urban mothers had poor nutritional status, often exacerbated by the lack of or limited childcare support and financial assistance during their work absences. Additionally, many South Asian countries implemented maternal and paternal leave policies and benefits; however, the implementation challenges reduced maternal access to these benefits and predominantly favored mothers working in formal sectors. Conclusions: This systematic review underscores the necessity to strengthen the employment benefit programs for working women in South Asian countries, especially for mothers working in informal sectors. The provision of childcare assistance and supplementary financial benefits to women employed in informal sectors is essential to improve the child nutritional outcomes.

## 1. Introduction

Globally, the prevalence of malnutrition in children aged birth to 12 years is drastically high, with 149 million children stunted, 37 million overweight, and 45 million children wasted [1]. Malnutrition remains a perennial challenge, predominantly affecting the most vulnerable groups, especially in the lowest economic strata [2,3]. Evidence suggests that children under five and school-aged children experience several forms of malnutrition, including stunting, wasting, underweight, overweight, obesity, and micronutrient deficiencies [4,5,6].

Malnutrition hinders growth and development and increases morbidity and mortality rates in children [7,8,9]. Inappropriate infant and child-feeding practices, socioeconomic conditions, food insecurity, inadequate access to care, and poor dietary resources are the key determinants of child undernutrition [10,11,12]. Additional factors linked to malnutrition are time-restricted mothers, insufficient family support, and the premature introduction of ready-to-eat foods [13,14]. On the other hand, adequate childcare improves child survival, growth, and development [15]. The provision of adequate nutrition and care constitutes the fundamental responsibilities of mothers. Multiple factors influence a mother’s capacity to care for her children, including education and knowledge, physical and mental health, workload, time available, autonomy utilizing family resources, and social support [16]. In South Asia, most women cease employment post-delivery owing to childcare responsibilities. Many women from economically weaker sections work predominantly in the informal sectors due to limited educational opportunities to supplement family income. Employment in the informal sectors lacks benefits, such as childcare assistance or maternity leave, which adversely impact the health of children [17].

A woman’s income increases food security, ensures nutrition security, and facilitates access to adequate childcare services that improve the child’s nutrition status [18,19]. The dual responsibilities of childcare and employment induce stress in working women [20]. Additionally, the reduced time mothers spend with their children due to employment impacts adequate childcare, resulting in poor nutritional outcomes in children [13,21]. The association between maternal employment and the child’s health is influenced by factors such as the nature of employment, the employer, and maternal income [22]. The challenges are exacerbated for mothers from underprivileged households who are working for a living. The conceptual framework illustrates the association between maternal employment and children’s nutritional status (Figure 1). A trade-off between a mother’s income and the time available for childcare and the impact on child nutritional outcomes remains inconclusive. Hence, the present review aims to examine evidence on the association between maternal employment and the nutritional status of children (up to twelve-year-old children) from South Asian countries. Additionally, we compile, document, and analyze the existing maternity, paternity, and childcare policies implemented in South Asian countries to draw meaningful policy implications and recommendations for women in the workforce across the region.

## 2. Materials and Methods

The present systematic review examined the association of maternal employment status with the nutritional status of children (both chronic malnutrition-stunting and other forms of malnutrition) up to twelve years old from South Asian countries. We used the Preferred Reporting Items for Systematic Reviews and Meta-analysis (PRISMA) guidelines to report the included studies and narratively synthesize the findings [24].

### 2.1. Information Sources

An electronic search was conducted in three databases, viz., PubMed, Science Direct, and Web of Science, for full-text research articles published in the English language between January 2011 and December 2024. The Scopus database was not used in this study due to limited access to full-text articles. Additionally, a grey literature search was conducted to collect information on maternity leave, paternity leave, and childcare provision policies in South Asian countries to summarize the existing policies.

According to World Bank data, female labour force participation rates in South Asian countries fluctuated consistently between 2005 and 2010. However, since 2011, these rates have remained relatively stable. Notably, a major source of employment for women in this region is the informal sector [25,26]. In 2011, the International Labour Organization (ILO) introduced the Domestic Workers Convention, 2011 (No. 189), which aimed to ensure that domestic workers received the same protections as other workers. Given the significance of this labour policy convention, this review focuses on studies published from January 2011 onwards [27].

### 2.2. Eligibility Criteria

The research articles were selected based on well-defined Population, Intervention, Comparison, Outcome, Study Design, and Time Frame (PICOST) criteria (Table 1).

### 2.3. Search Strategy

The search strategy was developed and pretested by a team of reviewers and experts using a combination of well-defined keywords and Boolean operators in PubMed, Science Direct, and Web of Science databases. The search strategy used to search the research articles include: [“children” OR “young children” OR “infant” OR “child” OR “toddler” OR “preschool children”] AND [“women” OR “mother” OR “maternal”] AND [ “maternal employment” OR “working status” OR “women, working” OR “working women” OR “working woman” OR “employment status” OR “occupational status” OR “labor force” OR “gainfully employed] AND [“Afghanistan” OR “ Bangladesh” OR “Bhutan” OR “India” OR “Maldives” OR “Nepal” OR “Pakistan” OR “ Sri Lanka”] AND “South Asia” AND [“nutritional status” OR “physical growth” OR “anthropometry” OR “wasting” OR “stunting” OR “underweight” OR “undernutrition” OR “malnutrition” OR “obesity” OR “overweight” OR “composite index of anthropometric failure” OR “coexisting forms of malnutrition OR MND’s OR micronutrient deficiencies”].

### 2.4. Study Selection Process

All searched articles were uploaded in COVIDENCE 2.0 systematic review software (Veritas Health Innovation, Melbourne, Australia) in PubMed file format from the PubMed database and RIS file format from the Science Direct and Web of Science databases, respectively [28]. All duplicate articles were removed, and two reviewers independently performed the title and abstract screening followed by the full-text screening to include the eligible studies for the review. In case of any discrepancy, expert advice was taken to make the final decision through consensus regarding the inclusion and exclusion of the study. We retained the full-text research articles published in the English language in the review. The final included research articles are presented in the PRISMA flow diagram (Figure 2).

### 2.5. Assessment of the Reporting Quality of Selected Observational Studies

We used the Strengthening the Reporting of Observational Studies in Epidemiology-Modified (STROBE-M) tool to assess the reporting quality of the included observational studies. The studies were graded as “excellent” (≥85), “good” (70 to <85), “fair” (50 to <70), and “poor” (<50) based on the total obtained score from two independent reviewers [29,30].

### 2.6. Data Extraction and Analysis

The data were extracted and documented in the Microsoft Excel spreadsheet in the following format: author(s), year, country, age, sample (n), data collection tools, maternal employment status, indicators of childhood nutritional status, and overall findings. All of the included studies were examined to evaluate the association between maternal employment status with: (1) stunting (height-for-age < −2 standard deviations (SD), (2) other forms of malnutrition include a composite index of anthropometric failure (CIAF), coexisting forms of malnutrition (CFM), wasting (weight-for-height < −2SD), underweight (weight-for-age < −2SD), mean Z-score for HAZ (height-for-age), mean Z-score for WAZ (weight-for-age), mean Z-score for WHZ (weight-for-height), mid-upper-arm-circumference, mean Z-score of mid-upper-arm-circumference-for-age (MUACZ), small arm (mid-upper-arm-circumference < 115 mm), thinness/wasting (Z-score of BMI-for-age < −2SD), overweight (Z-score of BMI-for-age > 2SD), mean Z-score for BMI-for-age (BAZ), height, and weight.

Additionally, the information regarding maternity leave, paternity leave, and childcare provision policies in South Asian countries was summarized under the country name, law/act, maternity leave duration (MLD), paternity leave duration (PLD), and childcare provision policy. The two reviewers independently verified the data for precision.

## 3. Results

The preliminary search yielded 10,247 research articles from three databases (Figure 2). Duplicates were removed (n = 7637), followed by title and abstract screening of 2610 studies. Subsequently, in accordance with the inclusion-exclusion criteria, full-text screening was performed for 36 articles, resulting in the inclusion of 19 eligible studies in the systematic review.

### 3.1. Characteristics of the Included Studies

Table 2 delineates the characteristics of the included studies. Among the 19 included studies, 4 originated from Bangladesh, 5 each from India and Nepal, 3 from Pakistan, and 2 from Sri Lanka. Figure 3 depicts the location of included studies.

The STROBE-M rating was utilized to assess the quality of observational studies, categorizing them as “excellent” (n = 1), “good” (n = 14), and “fair” (n = 4). The four studies classified as fair did not furnish critical details regarding participant recruitment, exposure to variables, duration of follow-up, and data collection methods. Studies missed addressing potential confounders, measures implemented to mitigate bias, the number of participants with incomplete or missing data for key variables, the transition of relative to absolute risk over time, the impact of funding on the study, or the availability of data online, as suggested by the STROBE-M assessment tool (Appendix A).

Out of 19 studies, 9 studies examined stunting (height-for-age < −2SD) as the outcome variable, making it the most often studied outcome variable [Bangladesh = 2, India = 2, Nepal = 3, Pakistan = 1, Sri Lanka = 1] [31,32,33,34,35,36,37,38,39]. Four investigations studied wasting (weight-for-height < −2SD) [India = 1, Nepal = 1, Pakistan = 2] [37,38,40,41], while three studies assessed underweight (weight-for-age < −2SD) [Nepal = 1, Pakistan = 1, Sri Lanka = 1] [38,39,42]. Two studies [Bangladesh = 1, Pakistan = 1] assessed the coexistence of malnutrition [43,44], whereas one Bangladeshi study examined the composite index of anthropometric failure as an outcome variable [45]. Two studies [Sri Lanka = 2] measured BMI-for-age of children as an outcome variable [39,46]. Two studies [India = 1, Nepal = 1] used low mid-upper-arm-circumference-for-age (MUAC-for-age < −2SD) as the outcome variable [37,47]. One study assessed the mean Z-score of height-for-age, weight-for-height, and mid-arm-circumference-for-age as outcome variables [Nepal = 1] [37]; another study [Nepal = 1] used small arm (MUAC < 115 mm) [45], and one [India = 1] height and weight as outcome variables [48]. No studies from Afghanistan, Bhutan, and Maldives were identified among the eight South Asian countries (Table 2).

**Table 2 nutrients-17-01059-t002:** Characteristics of the included research articles assessing the association of maternal employment status with the nutritional status of up to twelve years old children in studies selected from South Asian countries.

Sr. No.	Author	S	W	UW	HAZ	WAZ	WHZ	Body Mass Index		MUAC			Ht, Wt	CIAF	CFM	SR
								Thinness/ Wasting	OW	MUACZ	MUACZ < −2SD	MUAC < 115 mm				
1	Sumon et al. [43]	×	×	×	×	×	×	×	×	×	×	×	×	×	✓	G
2	Win et al. [31]	✓	×	×	×	×	×	×	×	×	×	×	×	×	×	G
3	Islam et al. [45]	×	×	×	×	×	×	×	×	×	×	×	×	✓	×	G
4	Huda et al. [32]	✓	×	×	×	×	×	×	×	×	×	×	×	×	×	F
5	Sk et al. [33]	✓	×	×	×	×	×	×	×	×	×	×	×	×	×	G
6	Ambadekar et al. [40]	×	✓	×	×	×	×	×	×	×	×	×	×	×	×	G
7	Tigga et al. [47]	×	×	×	×	×	×	×	×	×	✓	×	×	×	×	F
8	Deshmukh et al. [34]	✓	×	×	×	×	×	×	×	×	×	×	×	×	×	F
9	Yeleswarapu et al. [48]	×	×	×	×	×	×	×	×	×	×	×	✓	×	×	F
10	Hossain et al. [49]	×	×	×	×	×	×	×	×	×	×	✓	×	×	×	G
11	Sigdel et al. [42]	×	×	✓	×	×	×	×	×	×	×	×	×	×	×	G
12	Adhikari et al. [35]	✓	×	×	×	×	×	×	×	×	×	×	×	×	×	G
13	Budhathoki et al. [36]	✓	×	×	×	×	×	×	×	×	×	×	×	×	×	G
14	Brauner Otto et al. [37]	✓	✓	×	✓	×	✓	×	×	✓	✓	×	×	×	×	G
15	Khaliq et al. [44]	×	×	×	×	×	×	×	×	×	×	×	×	×	✓	E
16	Khan et al. [38]	✓	✓	✓	×	×	×	×	×	×	×	×	×	×	×	G
17	Iftikhar et al. [41]	×	✓	×	×	×	×	×	×	×	×	×	×	×	×	G
18	Shinsugi et al. [46]	×	×	×	×	×	×	✓	✓	×	×	×	×	×	×	G
19	Galgamuwa et al. [39]	✓	×	✓	×	×	×	✓	×	×	×	×	×	×	×	G

✓: Indicates parameter assessed; ×: Indicates parameter not assessed; S: Stunting; W: Wasting; UW: Underweight; OW: Overweight; HAZ: Height-for-age; WAZ: Weight-for-age; WHZ: Weight-for-height; MUAC: Mid-upper-arm-circumference; MUACZ: Mean Z-score of Mid-upper-arm-circumference-for-age; MUACZ < −2SD: Z-score of MUACZ < −2SD/Small arm/Wasting; Overweight: Z- score of BMI-for-age > 2SD; Thinness/wasting: BMI-for-age < −2SD; Ht: Height; Wt: Weight; CIAF: Composite Index of Anthropometric Failure; CFM: Coexisting Forms of Malnutrition; SR: STROBE M rating; G: Good; F: Fair; E: Excellent.

### 3.2. Association Between Maternal Employment and Stunting Among up to Twelve-Year-Old Children from South Asian Countries

Out of all (n = 19) studies, nine studies assessed the association between the mother’s working status and stunting (Bangladesh = 2, India = 2, Nepal = 3, Pakistan = 1, Sri Lanka = 1) (Table 3) [31,32,33,34,35,36,37,38,39]. Out of nine studies, four showed no association between maternal employment and stunting among up to twelve-year-old children in South Asian countries (Bangladesh = 1, India = 1, Pakistan = 1, and Sri Lanka = 1) (*p* > 0.05) [32,34,38,39]. Alternately, four studies in South Asian countries (Bangladesh = 1, India = 1, Nepal = 2) found an inverse relationship between mothers working and stunting in children up to 12 years old [31,33,35,37]. One Nepalese study reported a low risk of stunting in children whose mothers worked (AOR: 0.74, (0.59, 0.94); *p* < 0.05) [36].

Four studies reported a higher risk of stunting, with odds ratios ranging between 1.27 and 2.22 in children of employed mothers from Bangladesh, India, and Nepal. The Bangladeshi study found a 2.22 times significantly higher risk of stunting in children of employed mothers (AOR: 2.22, (1.16, 4.24); *p* < 0.05) [31]; and the Indian study reported a high risk of stunting in bidi worker mothers’ children (OR: 1.92, (1.18, 3.12); *p* < 0.05) [33]. Out of three Nepali studies, one revealed a high risk of stunting in children of employed mothers (AOR: 1.27, (1.01, 1.60); *p* < 0.05), while one found a high risk of stunting in children of labourers (*p* < 0.05) [35,37]. Collectively, the available evidence showed mixed trends in the association between maternal employment and childhood stunting, with 44.4% of the studies indicating a high risk for stunting and a similar percentage reporting no association.

**Table 3 nutrients-17-01059-t003:** Association between maternal employment and stunting in up to twelve-year-old children from South Asian countries.

Sr. No.	Author	Age (Months), Sample Size (N)	DCT	Nutritional Status of Children; Maternal Employment Status	Overall Result
1	Win et al. [31]	0–59, 346	SQ	Stunting; Currently working: AOR^a^ (95%CI): 2.22 (1.16–4.24); *p* < 0.05	A high risk of stunting was found in employed mothers’ children
2	Huda et al. [32]	<60, 7173	BDHS data 2004 and 2014	Stunting Currently working: ACI (C%): −0.003 (3%)	No association found between mothers’ employment and stunting in children
3	Sk et al. [33]	3–59, 731	SQ	Stunting; Bidi worker: AOR^b^ (95%CI): 1.92 (1.18–3.12); *p* < 0.05 Agricultural/manual worker: AOR (95%CI): 1.77 (0.55–5.66)	A high risk of stunting was found in working mothers’ children
4	Deshmukh et al. [34]	0–36, 990	SQ	Stunting; Housework: OR (95%CI): 1.5 (0.4–6.8) Service or business: OR: 1 Skilled work: OR (95%CI): 1.4 (0.2–10.5) Farm labourer: OR (95%CI): 2.3 (0.6–10.7) Unskilled manual labourer: OR (95%CI): 2.8 (0.8–13.4); *p* > 0.05	No association found between mothers’ employment and stunting
5	Adhikari et al. [35]	0–59, 9989	NDHS 2006, 2011 and 2016.	Stunting; Employed mother: AOR^c^ (95%CI): 1.27 (1.01–1.60); *p* < 0.05	A high risk of stunting was found in employed mothers’ children
6	Budhathoki et al. [36]	0–59, 16,086	NDHS 2001, 2006, 2011 and 2016	Stunting; Working mother: AOR (95%CI): 0.74 (0.59–0.94); *p* < 0.05	A low risk of stunting was found in employed mothers’ children
7	Brauner-Otto et al. [37]	3–60, 860	CVFS 2016 DER NHCs	Stunting; Ever: Wage labour: Coeff (SE): 0.443 (0.21); *p* < 0.05 Salary job: Coeff (SE): 0.174 (0.22) Own business: Coeff (SE): −0.011 (0.24) Before the child is born: Wage labour: Coeff (SE): 0.477 (0.24) *p* < 0.05 Salary job: Coeff (SE): 0.236 (0.26) Own business: Coeff (SE): 0.291 (0.30) 1st 1000 days: Wage labour: Coeff (SE): 0.354 (0.29) Salary job: Coeff (SE): −0.071 (0.31) Own business: Coeff (SE): −0.046 (0.31)	A high risk of stunting was found in children whose mothers worked as wage labourers
8	Khan et al. [38]	0–59, 3071	PDHS 2012–2013	Stunting; Working: AOR^d^ (95%CI): 0.92 (0.73–1.16); *p* > 0.05	No association found between mothers’ employment and stunting
9	Galgamuwa et al. [39]	12–<60, 547 (386 Excluded 11–15 yr.)	SQ	Stunting; Mother employed as labourers in tea plantations: OR (95%CI): 1.40 (0.68–2.68) *p* > 0.05	No association found between mothers’ employment and stunting

N: Sample size; DCT: Data collection tool; SQ: Structured questionnaire; BDHS: Bangladesh Demographic and Health Surveys; NDHS: Nepal Demographic and Health Surveys; ACI (C%): Absolute Contribution Index (Contribution%); OR: Odds Ratio; CI: Confidence Interval; Coeff (SE): Coefficient (Standard Error); CVFS: Chitwan Valley Family Study; DER: Demographic Event Registry; NHCs: Neighborhood History Calendars supplement; PDHS: Pakistan Demographic and Health Survey; AOR: Adjusted odds ratio; AOR^a^: Adjusted for child age, gender, birthweight, household income, maternal age and stature, parental education; AOR^b^: Child’s age, gender, birth order, birth interval and birth weight, duration of breastfeeding mothers’ age at birth, mother’s occupation and education, religion, caste, wealth index, place of residence; AOR^c^: Family size, headship of the household, caste/ethnicity, wealth quintile, place of residence, household food security status, access to drinking water and a toilet, mother’s age, mother’s years of education, number of living children, mother’s employment status, mother’s BMI and anemia status, age and sex of child, birth order, size of child at time of birth, and child anemia status; AOR^d^: Sex of child, age of child, child’s size at birth, antenatal clinic visits, recent diarrheal incidence, and breastfeeding status.

### 3.3. Association Between Maternal Employment Status and Other Anthropometric Indices Among up to Twelve-Year-Old Children from South Asian Countries

Thirteen of the nineteen studies examined the association between maternal employment status and various childhood anthropometric indices, including wasting/WHZ scores, underweight/WAZ scores, MUACZ scores, small-arm (mid-upper-arm-circumference < 115 mm), composite index of anthropometric failure (CIAF), coexisting forms of malnutrition (CFM), thinness/wasting (Z-score of BMI-for-age < −2SD), and overnutrition (overweight or obesity: Z-score of BMI-for-age > 2SD), as well as mean Z-score for BMI-for-age (BAZ), height, and weight (Table 4).

Among the 13 studies, 5 studies reported a negative association (India = 2, Nepal = 2, Sri Lanka = 1), 2 studies reported a positive association (Nepal = 1, Pakistan = 1), and 6 studies reported no association of maternal employment status with nutritional status of children (Bangladesh = 2, India = 1, Pakistan = 2, Sri Lanka = 1). Studies indicated that the predominant maternal occupation was unskilled labour, particularly daily wage earners, and the adverse impact of maternal employment status on the children’s nutritional status was primarily evident in daily wage labourer mothers.

Among the five studies that reported an association between maternal employment status and childhood nutritional status, an Indian study found an increased risk incidence of severe acute malnutrition among the offspring of employed women (OR: 2.4 (1.9, 3.1); *p* < 0.05) [40]. A separate Indian study found that children of unemployed mothers were significantly taller (*p* < 0.05) and heavier (*p* < 0.05) than children of employed mothers [48]. A Nepali study revealed that the risk of severe acute malnutrition was high in daily-wage labourer mothers’ children (OR: 2.15 (1.04, 4.65); *p* < 0.042) [49]. Similarly, another study from Nepal found poor nutritional status in daily wage-earner mothers’ children [37]. A Sri Lankan study reported a high prevalence of underweight (OR: 2.12 (1.05, 4.28); *p* < 0.037) and wasting (OR: 2.12 (1.05, 4.28); *p* < 0.037) in children whose mothers were employed as daily wage labourers in tea plantations [39]. Conversely, a Nepalese study revealed better nourishment among children of employed mothers than children of unemployed mothers (AOR: 4.29 (1.25, 14.78); *p* < 0.05) [42]. Similar findings were reported in a Pakistani study that found a low-risk coexistence of underweight with wasting in children of working mothers (AOR: 0.47 (0.23, 0.95); *p* < 0.05) [44].

The results indicated regional variations in the association between maternal employment and child nutritional status. Three studies from urban areas of South Asian countries reported poor nutritional outcomes. For example, a Bangladeshi study reported a high prevalence of stunting among children of urban working mothers, who were 63% employed in unskilled jobs [31]. An Indian study from urban slums highlighted poor nutritional outcomes with lower heights and weights of children compared to unemployed mothers [48]. Similarly, Sri Lankan children of employed mothers were marginally overweight and obese, although the association between maternal employment status and nutritional outcomes was not significant [46].

Overall, the findings showed mixed trends: 8/19 studies reported no association between maternal employment status and childhood nutritional status (Bangladesh = 3, India = 2, Pakistan = 2, Sri Lanka = 1) [33,34,38,41,43,45,46,47]; and another 8/19 studies presented a negative association (Bangladesh = 1, India = 3, Nepal = 3, Sri Lanka = 1) [31,33,35,37,39,40,48,49]. On the contrary, three studies (Nepal = 2; Pakistan = 1) suggested improved nutritional status for children of employed mothers [36,42,44]. To summarize, the current evidence indicated that the type of employment of mothers (i.e., skilled vs. unskilled) and regional variations determined the association between maternal employment status and childhood nutritional status.

**Table 4 nutrients-17-01059-t004:** Association between maternal employment status and other anthropometric indices among up to twelve-year-old children from South Asian countries.

Sr. No.	Author	Age (Months), Sample Size (N)	DCT	Nutritional Status of Children; Maternal Employment Status	Overall Result
1	Sumon et al. [43]	0–59, 7127	BDHS data 2017–18	CFM; Working mother: OR (95%CI): 1.03 (0.92–1.14); *p* > 0.05	CFM was not related to mothers’ working status
2	Islam et al. [45]	0–59, 6965	BDHS data 2014	CIAF; Working mother: AOR^a^ (95%CI): 1.3 (0.71–2.37); *p* > 0.05	CIAF was not related to mothers’ working status
3	Ambadekar et al. [40]	6–60, Cases: 737 Controls: 737	SQ	Wasting (WHZ < −3SD); Working mother: OR (95%CI): 2.4 (1.9–3.1); *p* < 0.05	The risk of wasting was high in working mothers’ children
4	Tigga et al. [47]	12–60, 1222	SQ	Wasting (MUACZ < −2SD); Working mother: OR (95%CI): 0.67 (0.53–0.85)	Wasting was not related to mothers’ working status
5	Yeleswarapu et al. [48]	24–60, 623	SQ	Weight (kg); EM’s children 2+ yr.: Mean ± SD 8.97 ± 0.67; 3+ yr.: Mean ± SD 10.2 ± 1.05; 4+ yr.: Mean ± SD 11.7 ± 1.46 UM’s children 2+ yr.: Mean ± SD 9.65 ± 1.09; 3+ yr.: Mean ± SD 11.1 ± 1.34; 4+ yr.: Mean ± SD 12.4 ± 1.49; *p* < 0.05 Height (cm); EM’s children 2+ yr.: Mean ± SD 75.4 ± 2.9; 3+ yr.: Mean ± SD 81.4 ± 4.5; 4+ yr.: Mean ± SD 89.3 ± 6.1 UM’s children; 2+ yr.: Mean ± SD 77 ± 4.3; 3+ yr.: Mean ± SD 83.7 ± 5.6; 4+ yr.: Mean ± SD 89.1 ± 6 *p* < 0.05	The height and weight of unemployed mother’s children were better than those of employed mothers’ children
6	Brauner-Otto et al. [37]	3–60, 860	CVFS 2016 DER NHCs	HAZ-M; Ever: Wage labour: EE (SE): −0.279 (0.11); *p* < 0.05 Salary job: EE (SE): −0.128 (0.11) Own business: EE (SE): −0.054 (0.12) Before the child is born: Wage labour: EE (SE): −0.232 (0.12) Salary job: EE (SE): −0.176 (0.13) Own business: EE (SE): −0.094 (0.15) 1st 1000 days: Wage labour: EE (SE): −0.031 (0.14) Salary job: EE (SE): −0.111 (0.16) Own business: EE (SE): 0.109 (0.16) WHZ-M; Ever: Wage labour: EE (SE): −0.289 (0.10); *p* < 0.05 Salary job: EE (SE): 0.061 (0.11) Own business: EE (SE): −0.033 (0.11) Before the child is born: Wage labour: EE (SE): −0.189 (0.11); *p* < 0.05 Salary job: EE (SE): 0.116 (0.12) Own business: EE (SE): −0.074 (0.14) 1st 1000 days: Wage labour: EE (SE): −0.255 (0.14) Salary job: EE (SE): −0.031 (0.15) Own business: EE (SE): 0.039 (0.15) MUACZ-M; Ever: Wage labour: EE (SE): −0.356 (0.08); *p* < 0.05 Salary job: EE (SE): −0.054 (0.08) Own business: EE (SE): −0.111 (0.09) Before the child is born: Wage labour: EE (SE): −0.243 (0.09); *p* < 0.05 Salary job: EE (SE): −0.009 (0.10) Own business: EE (SE): −0.182 (0.11) 1st 1000 days: Wage labour: EE (SE): −0.284 (0.11); *p* < 0.05 Salary job: EE (SE): −0.079 (0.12) Own business: EE (SE): −0.015 (0.12) Wasting (WHZ < −2SD); Ever: Wage labour: Coeff (SE): 0.265 (0.32) Salary job: Coeff (SE): 0.316 (0.36) Own business: Coeff (SE): 0.155 (0.38) Before the child is born: Wage labour: Coeff (SE): 0.255 (0.34) Salary job: Coeff (SE): 0.323 (0.39) Own business: Coeff (SE): 0.158 (0.47) 1st 1000 days: Wage labour: Coeff (SE): 0.033 (0.52) Salary job: Coeff (SE): −1.169 (1.06) Own business: Coeff (SE): −1.165 (1.06) MUACZ < −2SD; Ever: Wage labour: Coeff (SE): 1.283 (0.44); *p* < 0.05 Salary job: Coeff (SE): −0.271 (0.50) Own business: Coeff (SE): 0.684 (0.47) Before the child is born: Wage labour: Coeff (SE): 0.332 (0.45) Salary job: Coeff (SE): 0.599 (0.54) Own business: Coeff (SE): 0.291 (0.30) 1st 1000 days: Wage labour: Coeff (SE): 1.117 (0.56); *p* < 0.05 Salary job: Coeff (SE): −0.075 (0.68) Own business: Coeff (SE): −0.099 (0.80)	Significantly lower mean HAZ, WHZ, MUACZ score, and high risk of the small arm was found in children whose mother worked as a wage labourer
7	Sigdel et al. [42]	0–60, Cases: 93 Controls: 186	SQ	Underweight (WAZ < −2SD); Mother not earning: AOR^b^ (95%CI): 4.29 (1.25–14.78); *p* < 0.05 Mother Earning: 1	A lower risk of being underweight was found in children whose mothers were earning
8	Hossain et al. [49]	6–59, Cases: 128 Controls: 128	SQ	SAM (MUAC < 115 mm); Daily wage labourer: OR (95%CI): 2.15 (1.04–4.65); *p* < 0.05 Regular employee: OR (95%CI): 0.28 (0.12–0.58)	The risk of SAM was high in daily labourer mothers’ children than regular employed mothers’ children
9	Khaliq et al. [44]	0–59, 6168	PDHS data 2012–13 and 2017–18	Coexistence of Underweight with Wasting; Working mother: AOR^c^ (95%CI): 0.47 (0.23–0.95), *p* < 0.05 Coexistence of Underweight with Stunting Working mother: OR (95%CI): 0.95 (0.55–1.67) Coexistence of Underweight with Wasting and Stunting Working mother: OR (95%CI): 0.82 (0.44–1.55) Coexistence of Stunting with Overweight/Obesity Working mother: AOR^d^ (95%CI): 0.49 (0.31–0.79), *p* < 0.05	A lower risk of coexistence of underweight with wasting and stunting with overweight/obesity in employed mothers’ children
10	Khan et al. [38]	0–59; 3071	PDHS 2012–2013	Underweight (WAZ < −2SD); Working mother: AOR^e^ (95%CI): 0.96 (0.67–1.38); *p* > 0.05 Wasting (WHZ < −2SD) Working mother: AOR^e^ (95%CI): 0.72 (0.43–1.23); *p* > 0.05	No association found between mothers’ employment and underweight as well as wasting
11	Iftikhar et al. [41]	6–60; 340	SQ	Wasting (WHZ < −2SD); Employed mother: OR (95%CI): 1.37 (0.46–3.80) *p* > 0.05	No association found between mothers’ employment and wasting
12	Galgamuwa et al. [39]	12–<60, 547 (386 excluding > 12 yr)	SQ	Underweight (WAZ < −2SD); Mother employed as labourers in tea plantations: OR (95%CI): 2.12 (1.05–4.28); *p* < 0.05 Wasting (BAZ < −2SD) Mother employed as labourers in tea plantations: OR (95%CI): 2.12 (1.05–4.28); *p* < 0.05	A high risk of being underweight and wasting was found in employed mothers’ children
13	Shinsugi et al. [46]	60–120, 543	SQ	Thinness (BAZ < −2SD); Employed mother: AOR^f^ (95%CI): 0.64 (0.30–1.39) Overweight (BAZ > +1SD) Employed mother: AOR^f^ (95%CI): 0.72 (0.31–1.68), *p* > 0.05	No association found between mothers’ employment and thinness as well as overweight

CFM: Coexisting Form of Malnutrition; CIAF: Composite Index of Anthropometric Failure; CVFS: Chitwan Valley Family Study; DER: Demographic Event Registry; NHCs: Neighborhood History Calendars supplement; DCT: Data Collection Tool; SQ: Structure Questionnaire; SE: Standard Error; HAZ: Height-for-Age; WAZ: Weight-for-Age; WHZ: Weight-for-Height; SAM: Severe Acute Malnutrition; MUACZ: Mid-upper-arm-circumference-for-age; BAZ: BMI-for-Age; HAZ-M: Mean Z-Score of Height-for-Age; WHZ-M: Mean Z-Score of Weight-for-Height; MUACZ-M: Mean Z-Score of Mid-upper-arm-circumference-for-age; EE (SE) Effect Estimate (Standard Error); Coeff (SE): Coefficient (Standard Error); AOR: Adjusted Odds Ratio; COR: Crude Odds Ratio’s; SE: Standard Error; OR: Odds Ratio; CI: Confidence Interval; SE: Standard Error; EM’s: Employed Mother’s; UM’s: Unemployed Mother’s; AOR^a^: Child’s age, gender, mothers’ age at birth, Immunization status, mother’s occupation and education, religion, caste, wealth index, place of residence, Media exposure, delivery by C section, preceding birth interval, birth order, ANC care, size of child at birth, Mother’s BMI; AOR^b^: adjusted to child gender; AOR^c^: Adjusted for socioeconomic status, and maternal working status; AOR^d^: Adjusted for socioeconomic status, child age, maternal employment status, type of place of residence, and survey year; AOR^e^: Sex of child, age of child, child’s size at birth, antenatal clinic visits, recent diarrheal incidence, and breastfeeding status; AOR^f^: child sex, maternal education, maternal employment status, household equivalent income, maternal age, and maternal nutritional status.

### 3.4. Maternity Leave, Paternity Leave, and Childcare Policies in South Asian Countries

In South Asian countries, several laws/acts such as the Afghanistan Labour Law (2011), Bangladesh Labour Act (2011), Bhutan Regulations on Working Conditions (2012), India Maternity Benefit Act (2017), Maldives Employment Act (2008), Nepal Labour Act (2017), Pakistan Sindh Maternity Benefit Act (2018), and Sri Lanka Maternity Benefits Ordinance (2018) are enacted to assist working parents in the care of their children. These laws and acts primarily aimed to improve the nutritional status of children in South Asia. The literature indicates that nearly all South Asian countries have implemented the ILO’s maternity leave policy; however, among the eight South Asian countries, only India and Bangladesh offer full 14 weeks of maternity leave and childcare provisions for women employed in the formal sector. The countries Afghanistan, Bhutan, Maldives, Nepal, Pakistan, and Sri Lanka offer < 14 weeks of maternity leave (Table 5).

Paternity leave policy ensures that working fathers are granted leave to address their children’s healthcare requirements, and paternity leave policies are implemented solely by a limited number of nations, including Bhutan, Maldives, and Nepal. In the context of childcare facility policies for children of working mothers in South Asia, the policy has been implemented by countries including Bangladesh, India, and Nepal. According to the policy, childcare facilities must be provided for mothers of under-five children who are employed in any establishments with >30 female employees. The beneficiaries of the labour policies are the employees in the formal sector. Informal sectors employing women as domestic servants, street vendors, garbage pickers, construction workers, or in other informal occupations are ineligible for maternity leave, paternity leave, and childcare policies that are exclusively available to formal sector employees with children under six years of age; however, no policies exist for parents of children older than six years [17].

Numerous social security and welfare measures have been instituted in South Asian countries to assist informal workers, to enhance their livelihoods, and to mitigate their vulnerability. The Bangladesh National Social Security Strategy offers health insurance, social security, and comprehensive skill training to vulnerable groups [50]. India has enacted several policies and legislations to support informal workers, including Ayushman Bharat, the Mahatma Gandhi National Rural Employment Guarantee Act, the National Pension Scheme, and Pradhan Mantri Shram Yogi Maan-Dhan [51]. The Social Security Fund and National Health Insurance Program in Nepal are established to support workers in informal sectors [52]. The Samrudhi Program of Sri Lanka provides financial and social welfare service to informal workers [53]. Despite extensive efforts by governments throughout South Asia, substantial challenges remain in guaranteeing the effectiveness and accessibility of these programs, particularly for the most disadvantaged groups.

Several other employment benefits exist across South Asia, requiring compliance with the National Labour legislation, adherence to income tax rules, provision of social protection, and entitlement to specific benefits, including regulations on layoffs, severance pay, maternity leave, and other types of leave. However, it pertains solely to the employees of formal sectors. In contrast, informal employment does not confer such benefits to the employees. In the absence of social protection measures such as paid maternity, paternity, and parental leave; child and family benefits; and free or significantly subsidized childcare, working parents and families in the informal economy face an added financial burden in child-rearing [17]. In India, women employed as domestic workers often serve as the primary earners for their families and resume work shortly after childbirth. The absence of maternity benefits for women in the informal sector endangers maternal health and adversely impacts their well-being and the development of their child [54].

## 4. Discussion

The present systematic review, to our knowledge, is the first of its kind that comprehensively examines the association between maternal employment and the nutritional status of children in conjunction with the existing policies of South Asian countries. Our findings show mixed results on the association between maternal employment and childhood nutritional status—with some beneficial effects, no effects, and adverse effects. Additionally, the association was determined by the nature or type of jobs, workplace environment, availability of childcare facilities, and regional variations.

Maternal employment contributes to household resources; household wealth plays a more significant role in reducing child undernutrition than maternal employment alone [55]. The association between mothers’ employment and children’s nutritional outcomes is multifaceted, with income and caregiving being a key determinant. Unemployed mothers’ children may be better nourished than employed mothers’ children. However, these dynamics may reverse if the unemployed mother comes from an underprivileged household (Figure 1) [56].

The adverse effects on nutritional status were predominantly contextual, with mothers employed in low-paying jobs in informal or unskilled sectors such as wage labour, agriculture, and small-scale industries. Recently, the ILO reported that in South Asia, predominantly women are employed in the agricultural sector (57.6%), followed by the industrial section (12.8%), with 9.8% of them engaged in low-value-added jobs. Furthermore, 6.3%, 3.7%, and 13% of women in South Asia work in trade, construction, and other sectors, respectively, while only 6.5% are employed in education. Despite a significant increase in women’s labour force in South Asia, many women are engaged in poor value-added or informal sectors of employment. Women working in the informal sectors largely belong to poor, underprivileged communities where balancing work and caregiving responsibilities is a challenge.

Employment in informal sectors is unstable and lacks basic health, welfare services, and social security [57]. Although maternity leave, paternity leave, and childcare policies are implemented in many countries, the benefits are available in formal work sectors for children under six years, and none are available for mothers in informal sectors. A review of Global South Asian countries revealed that barely one percent of working women in low-income communities have access to adequate childcare facilities; it is either due to the unavailability of quality childcare services or the exorbitant cost of quality childcare services [58]. According to a 2015 survey of United Nations women, working women from low-income communities often bring their children to the workplace in the absence of a substitute caregiver [59].

Findings from our review endorse that the nature of employment or job type and the availability of childcare support services in the mother’s absence significantly impact children’s nutritional status. Poor child nutritional outcomes were attributed to inadequate childcare support [31], unsanitary and infection-prone environments of workplaces exposing children to health risks, as many mothers bring their children to work, limited time for childcare, and greater reliance on junk foods—a scenario that is more evident in urban areas [48]. The present systematic review highlights the importance of caregiver substitutes for addressing the children’s nutritional needs. The caregiver substitute can be the family member, hence understanding the family structure and the role of family members in caregiving practices is necessary; however, this information is lacking in the included studies of the present systematic review. This review underscores the intricate and complex association between maternal employment and children’s nutritional status with the interplay of multiple factors such as mothers’ income, work environment, and nature of employment that determines the availability of resources for childcare, and in turn the nutritional outcomes.

### Strengths and Limitations

Our review is the first to indicate the possible association of maternal employment status with the nutritional status of up to twelve-year-old children from South Asian countries. The strength of this review is the adoption of a well-defined search strategy that covered three databases and robust methodology to synthesize this systematic review. All of the included studies used standard tools to assess various parameters of malnutrition in children and had a larger sample size. Furthermore, this review also explored the existing maternal leave, paternal leave, and childcare policies from South Asian countries. The current systematic review is limited by its inclusion of only English-language studies, non-inclusion of the Scopus database, and the reliance on grey literature to compile the maternity leave, paternity leave, and childcare provision policies in South Asia. Additionally, there was paucity of information on the family structures and the role of marriage and fathers in the employment dynamics in the included primary studies.

## 5. Conclusions

This review concludes that the association between maternal employment and child nutritional outcomes in South Asia is complex and context-dependent. Policymakers and administrators should address the complex challenges of women employed in informal sectors through the implementation of employment legislation, provision of childcare facilities, and the establishment of social security measures to support mothers and improve child nutritional outcomes.

Nutritional outcomes of working mothers’ children can be further safeguarded by sponsoring childcare solutions in the community, integrating nutrition programs within workplaces, and ensuring flexible working options for breastfeeding mothers. Social protection, financial assistance, and targeted nutrition support should be provided to all informal sector’s working mothers and their children. To mitigate the adverse effect of maternal employment on the nutritional status of children, a multisectoral approach can be adopted wherein the NGOs and government jointly work to provide childcare support, food subsidies, and financial assistance to women working in the informal sectors.

## Figures and Tables

**Figure 1 nutrients-17-01059-f001:**
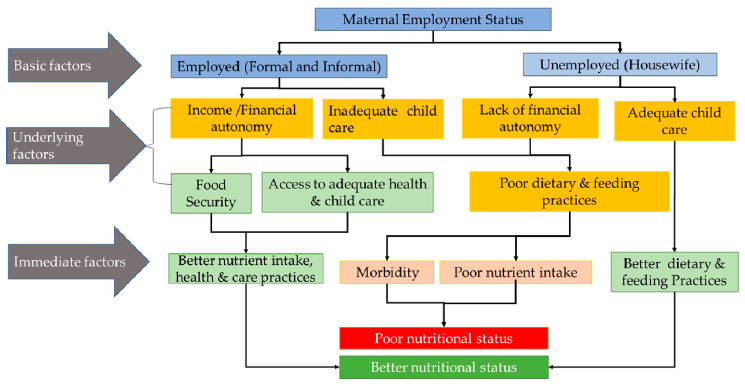
Conceptual framework illustrating the association between maternal employment and children’s nutritional status (adapted from UNICEF Conceptual Framework, 2020) [23].

**Figure 2 nutrients-17-01059-f002:**
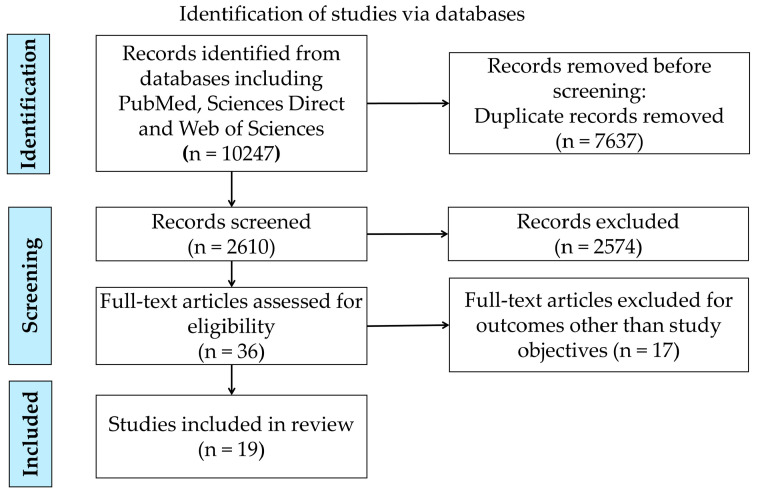
Preferred Reporting Items for Systematic Reviews and Meta-analysis (PRISMA) flow diagram showing the research article selection process.

**Figure 3 nutrients-17-01059-f003:**
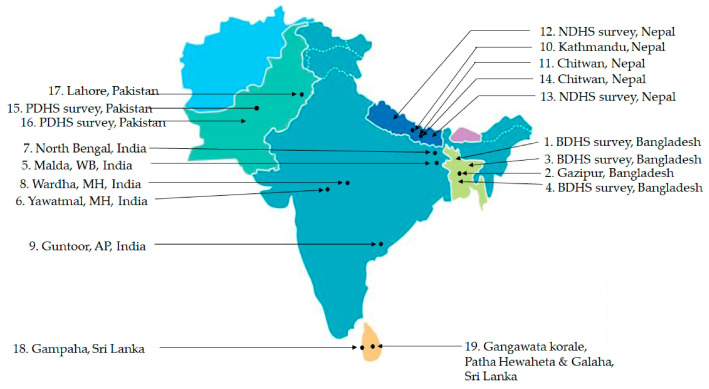
Distribution of included studies originated from South Asia in the present review.

**Table 1 nutrients-17-01059-t001:** Population, Intervention, Comparator, Outcome, Study Design, and Time Frame (PICOST) criteria for inclusion and exclusion of the research articles in this review.

Parameters	Inclusion Criteria	Exclusion Criteria
Population	Children up to 12 years of age	Children older than 12 years of age
Intervention	No intervention-studies that examined the association between maternal employment and the nutritional status of children	All other studies
Comparison	South Asian countries (Afghanistan, Bangladesh, Bhutan, India, Maldives, Nepal, Pakistan, and Sri Lanka)	Countries other than South Asian countries
Outcome	Malnutrition (stunting/HAZ score, wasting/WHZ scores, underweight/WAZ scores, MUACZ scores, small arm (mid-upper-arm-circumference < 115 mm) composite index of anthropometric failure (CIAF), coexisting forms of malnutrition (CFM), thinness/wasting (Z-score of BMI-for-age < −2SD), Overnutrition (overweight or obesity: Z score of BMI-for-age > 2SD); mean Z-score for BMI-for-age (BAZ), height, and weight	Studies that reported other outcome including assessment of micronutrients in children under 12 years of age
Study Design	Observational study (cross-sectional study, case control, and cohort study)	Experimental study designs (randomized controlled trial and quasi experimental design), opinion articles and editorials
Time Frame	January 2011 to December 2024	Studies published before January 2011

**Table 5 nutrients-17-01059-t005:** Maternity leave, paternity leave, and childcare provision policies in South Asian countries.

Sr. No.	Country	Law/Act (Year)	MLD (Weeks)	PLD (Days)	Childcare Provision Policy	
					Available	Beneficiary
1	Afghanistan	Labour Law (2011)	13	No	No	NA
2	Bangladesh	Labour Act (2011)	16	No	Yes	* Under six-year-old children
3	Bhutan	Regulations on working conditions (2012)	8	Yes (five)	INA	INA
4	India	Maternity Benefit Act (2017)	26	No	Yes	** Under six-year-old children
5	Maldives	Employment Act (2008)	8.6	Yes (three)	INA	INA
6	Nepal	Labour Act (2017)	7.4	Yes (15)	Yes	*** Under five-year-old children
7	Pakistan	Sindh Maternity Benefit Act (2018)	12	No	No	NA
8	Sri Lanka	Maternity Benefits Ordinance (2018)	12	No	No	NA

MLD: Maternity Leave Duration. PLD: Paternity Leave Duration. INA: Information Not Available. NA: Not Applicable. * Under six-year-old children whose mother is in formal employment and the establishment has more than 40 women workers. ** Under six-year-old children whose mothers are employed in the formal sector and the establishment has more than 30 female employees. *** Under five-year-old children whose mother is in formal employment.

## Data Availability

Data are contained within the article or Appendix A.

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
