# Peer review of "Work, Motherhood, and Nutrition: Investigating the Association of Maternal Employment on Child Nutritional Status in South Asia—A Systematic Review"

_nutrients, 2025, doi:10.3390/nu17061059_

Round 1
Reviewer 1 Report
Comments and Suggestions for Authors
This review constitutes a considerable effort on the part of the authors. They have carefully and thoughtfully interpreted these somewhat confusing results. I have only a few comments.
Introduction
Lines 70-72: The final sentence of the Introduction is not a complete sentence.
Figure 1: The conclusion of the Abstract heavily draws attention to the disadvantages of the informal work sector. However, this is not part of Figure 1. One simple solution would be to change the label “Employed” to “Employed (formal and informal)”.
Methods
Table 1: The search strategy included such words as “working status” and “employment status”. However, in Table 1 there is no mention of these employment Interventions.
Results
The role of fathers and of marital status. Most of the reviewed studies did not control for marital status and the word “married, and marriage”, were not found in the manuscript and “father” was found once in the policy section. Yet the children of unemployed women generally fared well. This is puzzling. Would the authors please address the role of marriage and fathers in these employment dynamics. It might be that a lack of information of family structures is a limitation of this study.
Author Response
Respected Sir,
Thank you for your suggestion.
Please find the attached file.

Reviewer 2 Report
Comments and Suggestions for Authors
paper title - Work, Motherhood, and Nutrition: Investigating the Association of Maternal Employment on Child Nutritional Status in South Asia—A Systematic Review
here are some comments, questions, and suggestions
in the abstract - would suggest for the second sentence - after ... a variety of social and economic conditions.
to add something like - Understadning the social and economic conditions would ......
- so as to make a better connection or significant of the review
why 2011 to 2024?
why these databases - PubMed, WOS, ScienceDirect? how about Scopus?
these should be noted later on in the methodology and limitations of the study
introduction is informative and concise
would help to reinterate the objective of the study preferably at the end of the introduction
material and methods
should note why up to 12 years old?
as noted earlier, in the information sources shold note why these specific databases
results and discussions seems adequate
just a note, are there any differences of the employment situation after enachment of certain government policy
is there a possibility of providing a chart or graph of the common variables as reflected within the studies against a timeline,
would help in visualizing the trends for instance in salary, nutritional status, .....
is it also possible to provide a certain norm value as a gauge of comparison for the various indicators, for instance, in the discussion, the authors could
note that OECD or UN standards.... for more clarity
conclusion can be expanded, what now?
what are some practical implications?
Author Response
Respected Ma'am,
Thank you for your suggestion.
Please find the attached file.

Reviewer 3 Report
Comments and Suggestions for Authors
There is value to perform systematic reviews on United Nations regions specific the socio-economic associations with pediatric outcome, including nutritional status. In essence, it seems that for the region analyses (South Asia), income seems to be more relevant than being employed (reflected in the unskilled, verus formal sectors). In essence, financial autonomy is likely better reflected by income, and not by employment status in itself. Could you retrieve information on the eg monthly revenues, weighted to the national references to quantify this, or does the literature not provide sufficient information to do so ?
Introduction: first lines, do you focus on the nutritional status in the first 5 years of life, or rather up to the age of 12 years ? Recommend to be consistent.
The title somewhat suggest a focus on South Asia, a United Nations regions, while this is subsequently narrowed to 5 countries. Why have you made this decision, and perhaps the title needs to be revised to eg 5 selected countries in South Asia (cf abstract), but the search strategy mentions more countries ? In essence, unclear, and therefore needs additional considerations.
Author Response

(The authors gave the same response as above.)
